# A New Decision Support System for Analyzing Factors of Tornado Related Deaths in Bangladesh



**Fahim Sufi** [1,*] , **Edris Alam** [2,3] **and Musleh Alsulami** [4]

1   Federal Government, Melbourne, VIC 3000, Australia
2   Faculty of Resilience, Rabdan Academy, P.O. Box 114646, Abu Dhabi 22401, United Arab Emirates;
    ealam@ra.ac.ae
3   Department of Geography and Environmental Studies, University of Chittagong,
    Chittagong 4331, Bangladesh
4   Information Systems Department, Umm Al-Qura University (UQU), Makkah 24382, Saudi Arabia;
    mhsulami@uqu.edu.sa
*   Correspondence: research@fahimsufi.com

**Abstract:** Tropical cyclones devastate large areas, take numerous lives and damage extensive property in Bangladesh. Research on landfalling tropical cyclones affecting Bangladesh has primarily focused on events occurring since AD1960 with limited work examining earlier historical records. We rectify this gap by developing a new Tornado catalogue that include present and past records of Tornados across Bangladesh maximizing use of available sources. Within this new Tornado database, 119 records were captured starting from 1838 till 2020 causing 8735 deaths and 97,868 injuries leaving more than 102,776 people affected in total. Moreover, using this new Tornado data, we developed an end-to-end system that allows a user to explore and analyze the full range of Tornado data on multiple scenarios. The user of this new system can select a date range or search a particular location, and then, all the Tornado information along with Artificial Intelligence (AI) based insights within that selected scope would be dynamically presented in a range of devices including iOS, Android, and Windows. Using a set of interactive maps, charts, graphs, and visualizations the user would have a comprehensive understanding of the historical records of Tornados, Cyclones and associated landfalls with detailed data distributions and statistics.

**Keywords:** logistic regression; AI based Tornado analysis; decision support system; mobile application

## 1. Introduction

Tornados are natural phenomenon that has adverse effect on human life, infrastructure, society, and economy [1,2]. For example, Hurricane Ida caused 64.5 billion US dollar of damages and 96 fatalities with power outages, landslides, and flash floods in just 4 days between 8 August 2021 and 1 September 2021 [3]. The adverse effects of Tornadoes and other disasters could be substantially reduced by research & development of early warning systems with Artificial Intelligence (AI) driven insights [4]. Even with the significant role of AI into Tornado research, there have been only few studies reported on this subject [5–7]. In [5], Artificial Neural Network (ANN) was used to predict properties damaged by Tornadoes. Researchers in [6] utilized Deep Neural Network (DNN) on thousands of imagery data for estimating damages caused by Tornadoes. Study in [7], reported Machine Learning (ML) based Tornado prediction algorithm used on a dataset of 10,816 events. All these studies, deployed AI and ML based algorithms on a large set of data [5–7] and obtained AI driven insights. Hence, if we wanted to obtain AI driven insights for Bangladesh, the prime issue comes down to scarcity of historical Tornado event data on which AI or ML algorithm could operate.

Previous research in Tornado, cyclone and landslides focused on a limited set of records in Bangladesh [1,8–17]. If we wanted apply AI and ML algorithms to generate

insights from a larger Tornado event set, then we needed to generate an aggregated Tornado event data for Bangladesh. In this paper, we reported a new Tornado database containing 119 records by aggregating Tornado events reported in earlier studies, local newspapers, TV channels, and magazines in Bangladesh [1,8–17]. Our new aggregated Tornado dataset reported Tornadoes in Bangladesh from 1838 till 2020 causing 8735 deaths and 97,868 injuries. This new database reveals that more than 102,776 people were affected by Tornado in Bangladesh. Within this database, the first cyclone recorded with geographic location was on 31 March 1875 at Mymensingh. The highest number of injuries took place at Jamalpur, Tangail causing 30,000 injuries on Monday, 13 May 1996. On the other hand, the highest number of deaths were caused by the cyclone at Dhaka on Monday 14 April 1969, causing 922 deaths. These data have been made publicly available in GitHub Source repository with under Creative Common license [18].

After generating the new Tornado dataset for Bangladesh, we used AI based regressions (i.e., both linear regression and logistic regression) to find out factors behind Tornado related deaths from these historical Tornado events. Our novel approach demonstrated the intricate relationships of various Tornado related parameters like speed, number of injuries, number of affected to the number of Tornado related deaths. The user of the proposed system could select a particular region of Bangladesh, and immediately our innovative algorithms would execute linear and logistic regression on the selected set of data to identify a range of factors responsible higher number of Tornado related deaths with detailed correlation factors. To visualize and analyze the Tornado data reported in this paper, we have also created a new AI based interactive system using Microsoft Power BI [19] for the users of our system. The interactive dashboard can be publicly accessed by any users using the web link at [20].

The new Tornado data presented within this paper along with the new dashboard system, would allow any strategic decision maker to make evidence-based policy decisions concerning Tornado events in Bangladesh. Therefore, the decision support system, presented in this paper would support end-to-end data-driven decision making as demonstrated by our previous research and literatures [2,21–29].

In Summary, following are the core contributions of this paper:

- This paper reported a new Tornado dataset for Bangladesh containing 119 records from 1838 till 2020 causing 8735 deaths, 97,868 injuries, and more than 102,776 people affected in total
- This study described the first AI driven solution for analyzing Tornado events in Bangladesh
- This paper demonstrated the applicability of Linear Regression and Logistic Regression for identifying the factors responsible for Tornado related deaths
- This paper depicted a fully deployed practical solution for disaster strategist that can be used in ubiquitous platforms (i.e., iOS, Android, and Windows)

## 2. Materials and Methods

The Tornado data presented in this study, were accumulated, and aggregated from a wide range of sources starting from local Newspapers (historical), journals and conference proceedings. The compiled Tornado data were subsequently cleaned and transformed before modelling. Secondly, data modelling using the best practice [30] was performed. Then, the data was visualized and analyzed using analytical visuals [31] and algorithms. Details of these analysis is portrayed within Section 2.4 of this paper. Figure 1 demonstrates the step-by-step of analyzing the Tornado data that is presented in this paper.

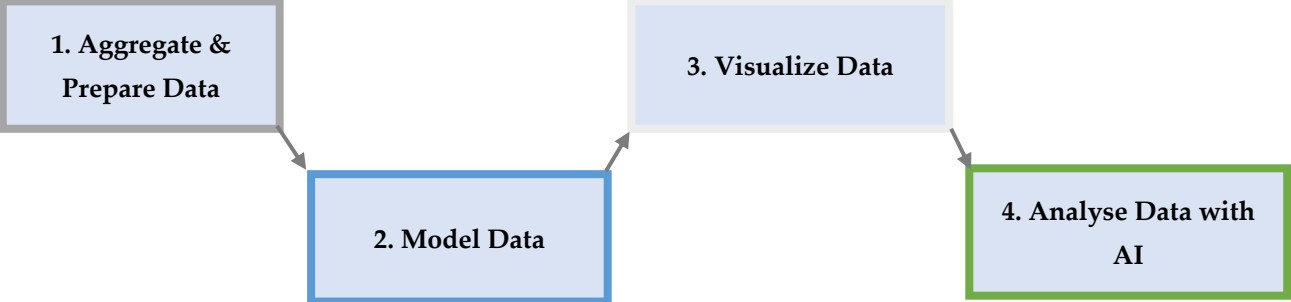

**Figure 1.** High level methodology of AI insight system for analyzing Tornadoes in Bangladesh.

*2.1. Aggregate and Prepare Data*

Data can be sourced from one or more sources. These sources can be of multiple types ranging from Online Databases, Websites, Excel Files, Flat Files, Web Based Application Programming Interfaces (APIs) or even PDF files. After identifying the data sources, data is obtained with data integration tools like SQL Server Integration Services (SSIS), Power BI Query Editor, Oracle data integrator, Tibco Pervasive Integration etc. These data integration tools facilitate Export, Transform, Load (ETL) process that obtains data from many different sources into a data warehouse. Specialized programming languages like Mashup (M) Language is used for data transformations and data cleansing as demonstrated in [2,21,22,26].

Data transformation and data cleansing is referred as "data preparation", since data needs to be transformed into right format before data can be modelled or analyzed. Within the scope of this paper, our accumulated and aggregated data were first compiled in Microsoft Excel format (i.e., .xlsx file extension). Using Microsoft Power BI's [19], Power Query Editor, the excel data was consumed and transformed. From the transformed data, the feature attributes of Tornado were better understood. Using Power Query Editor, various statistics about the attribute along with data distributions were produced. By default, the Microsoft Power Query Editor profiles the columns (i.e., produce column statistics) for top 1000 rows. However, for this study, we explicitly selected the option for column profiling based on the entire dataset to obtain comprehensive data statistics, as demonstrated in Table 1. Table 1 shows the detailed statistics of the presented Tornado data, including the name of the attribute, attribute type (i.e., data type), attribute statistics (showing how many empty values, how many distinct values, how many unique values etc.), and attribute distribution. The First row of Table 1 corresponds to event number (i.e., the unique identifier of the event). This event number is the mandatory ID of the event. There were 119 unique values corresponding to 119 Tornado records. In should be noted that the 4th row of Table 1 corresponds to location of landfall. This attribute (i.e., location of landfall) refers to the location of landfall initiated by Tornado related flash floods. Understanding the statistics for Tornado Feature Attribute are crucial before proceeding to the next steps of the methodology, namely Model Data, Visualize Data, and Analysis.

**Table 1.** Tornado attribute, data type & data distribution.

| Attribute Name | Attribute Type | Attribute Statistics | | | | Attribute Distribution |
| --- | --- | --- | --- | --- | --- | --- |
| | | Valid | Empty | Distinct | Unique | |
| 1. Event Number (e.g., 1, 2, 3 . . . 119) | Integer | 119 | 0 | 119 | 119 | ● Valid 100% ● Error 0% ● Empty 0% <br> 119 distinct, 119 unique |

**Table 1.** *Cont.*

| Attribute Name | Attribute Type | Attribute Statistics | | | | Attribute Distribution |
|---|---|---|---|---|---|---|
| | | **Valid** | **Empty** | **Distinct** | **Unique** | |
| 2. Date (e.g., 08/04/1838, 29/04/1972, 18/05/1991, 10/10/2007, ... ) | Date | 119 | 0 | 111 | 104 | Valid 100% / Error 0% / Empty 0% — 111 distinct, 104 unique |
| 3. District (e.g., Tongi, Bogra, Sirajgonj, Jessore ... ) | Text | 95 | 24 | 61 | 39 | Valid 80% / Error 0% / Empty 20% — 61 distinct, 39 unique |
| 4. Location of landfall (e.g., Bhakua and Haripur, Gounadi, Laksham, ... ) | Text | 48 | 71 | 47 | 44 | Valid 40% / Error 0% / Empty 60% — 47 distinct, 44 unique |
| 5. Area (in Square KM) (e.g., 51, 21, 75, 207, ... ) | Integer | 15 | 104 | 16 | 15 | Valid 13% / Error 0% / Empty 87% — 16 distinct, 15 unique |
| 6. Physical parameters speed (KM/Hour) (e.g., 115, 250, 322, ... ) | Integer | 18 | 101 | 16 | 13 | Valid 15% / Error 0% / Empty 85% — 16 distinct, 13 unique |
| 7. Physical parameters Duration (in Minutes) (e.g., 4, 5, 8, ... ) | Integer | 4 | 115 | 5 | 4 | Valid 3% / Error 0% / Empty 97% — 5 distinct, 4 unique |
| 8. Effect of Tornadoes—Death (in Person counts) (e.g., 1, 2, 15, 56, 681, ... ) | Integer | 103 | 16 | 58 | 38 | Valid 87% / Error 0% / Empty 13% — 58 distinct, 38 unique |

**Table 1.** *Cont.*

| Attribute Name | Attribute Type | Attribute Statistics | | | | Attribute Distribution |
|---|---|---|---|---|---|---|
| | | Valid | Empty | Distinct | Unique | |
| 9. Effect of Tornadoes—Injuries (in Person counts) (e.g., 35, 150, 200, 1000, … ) | Integer | 43 | 76 | 32 | 23 | Valid 36% / Error 0% / Empty 64% — 32 distinct, 23 unique |
| 10. Effect of Tornadoes—Affected (in Person counts) (e.g., 800, 1000, 5200, … ) | Integer | 13 | 106 | 11 | 8 | Valid 11% / Error 0% / Empty 89% — 11 distinct, 8 unique |
| 11. Effect of Tornadoes—Damage (in USD Millions) (e.g., 2, 10, 40–45, … ) | Integer | 15 | 104 | 10 | 6 | Valid 13% / Error 0% / Empty 87% — 10 distinct, 6 unique |
| 12. Key Comments (e.g., The train derailed in the northern district of jamalpur) | Text | 18 | 101 | 20 | 18 | Valid 15% / Error 0% / Empty 85% — 20 distinct, 18 unique |

We worked on a tabular dataset of 119 records and 12 Attributes. The Tornado Data Attributes can be grouped into 4 major categories: Definition, Categorization, Effect and Other. The Definition defines the Tornado in terms of Identifier, When (i.e., Date attribute) and Where (i.e., the location/district). The categorization criteria include attributes such as area in Sq.KM, speed, and duration of the Tornado. The Effect category hosts the most important attributes for conducting research and it include number of deaths and injuries that was caused by the Tornado. Moreover, it includes the number of people affected and the damage caused by the Tornado. Lastly, other category includes Location of Landfall and Key comments. Figure 2 shows classifications of the 12 attributes of Tornado data.

*2.2. Model Data*

Data modelling is the most important stage within the process of Generating Data Driven Insights. If data modelling is done correctly, the data analysis can produce powerful insights with minimum delay. During this stage, relationships among different sets of data is drawn with the right cardinality. As seen from Figure 3, the data obtained for this paper were arranged in a Star Schema [30], where the main factual data resides in the Center (i.e., referred in Figure 3 as Tornado Data). The central Tornado Data (i.e., fact table) are linked with two other dimensions (i.e., District and Date). This arrangement allows one-way filtering of the fact table (i.e., Tornado Data) by District and Date. The main benefit of start schema over other data modelling techniques (e.g., flattened table, snowflake etc.) is faster and provide more accurate results during data analysis [30].

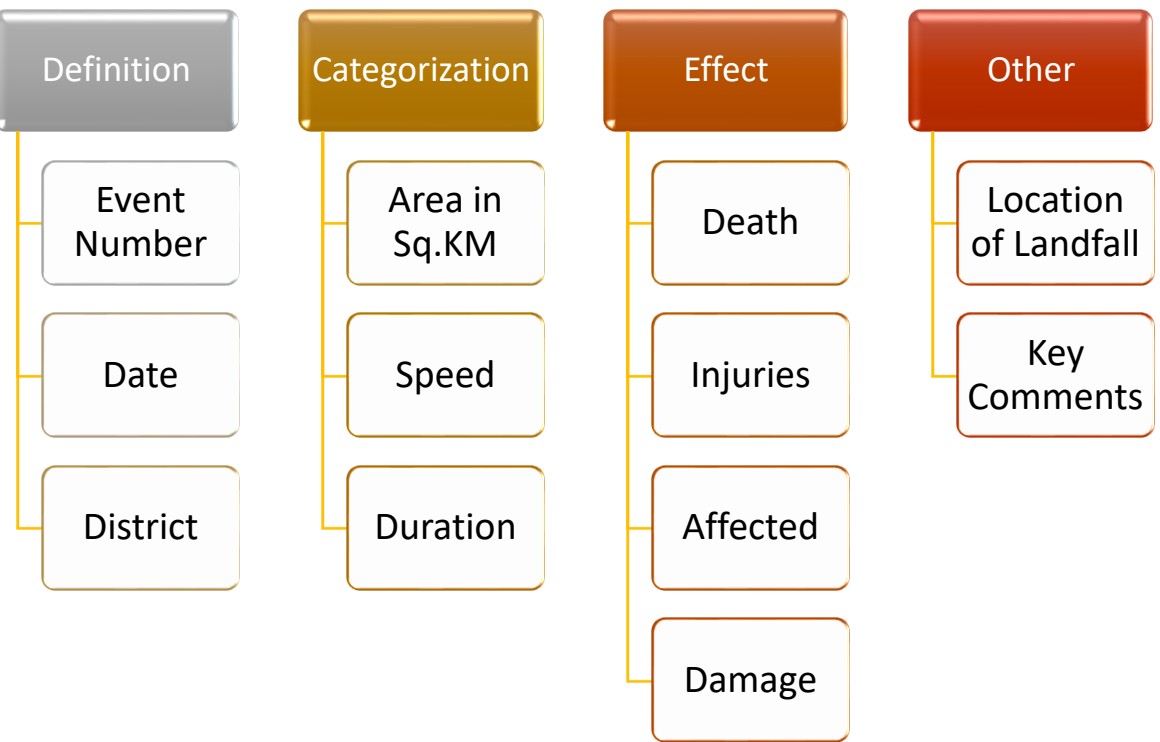

**Figure 2.** Classifying the 12 attribute of Tornado data.

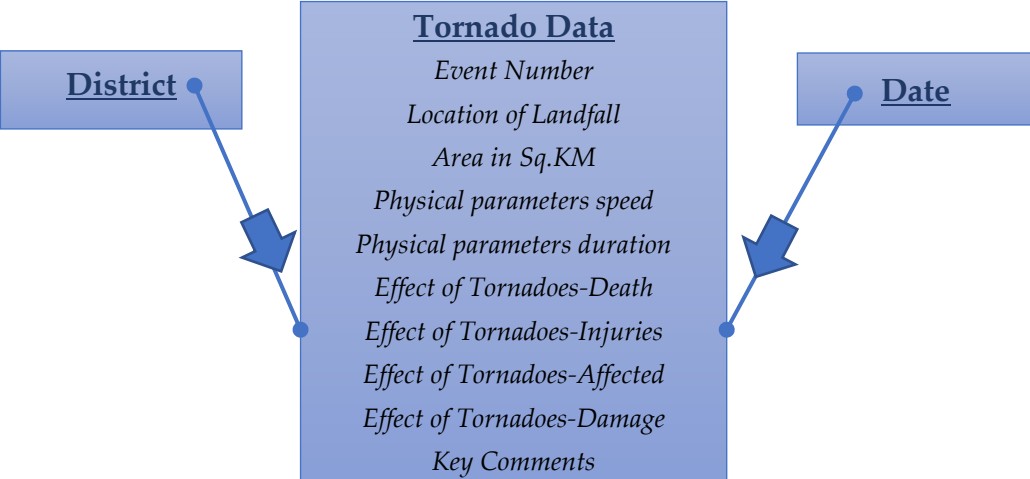

**Figure 3.** Data modelling of Tornado data.

*2.3. Visualize Data*

Once the data model was completed, we used district and date information to filter the Tornado data. A wide range of visualizations like slicer, ESRI Maps, Bar charts, Line Charts, Pie Charts, and data cards were used as seen from Figures 4 and 5. In Figure 5, Violin charts were used for data exploration and understanding of data statistics and distribution [31]. Unlike box plot, Violin charts can show data distributions on top of all the other statistics supported by box plots [31]. Changing the values for filter attributes like Date and District, filters the fact table (i.e., Cyclone Data), which in turns changes the visualizations like Violin charts (displayed in Figures 6 and 7). Case 1, Case2, and Case 3 as depicted in Figures 5–7 demonstrate the interactive nature of the presented system. With the change of input parameters, the data distributions are calculated on dynamically. This interactive visualization is publicly accessible with full analytical features at [20].

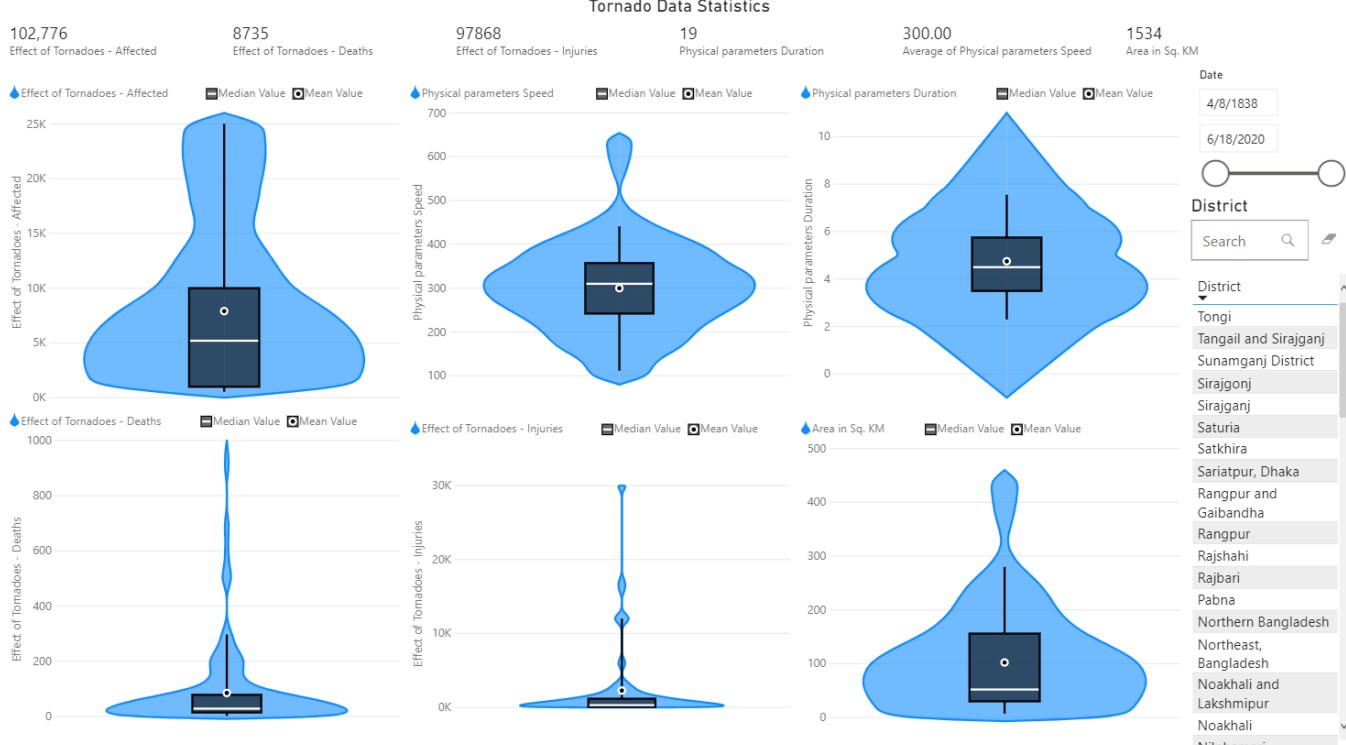

**Figure 4.** Dashboard for data exploration and analysis.

**Figure 5.** Dashboard for data statistics, distribution & analysis of key numeric attributes (Case 1).

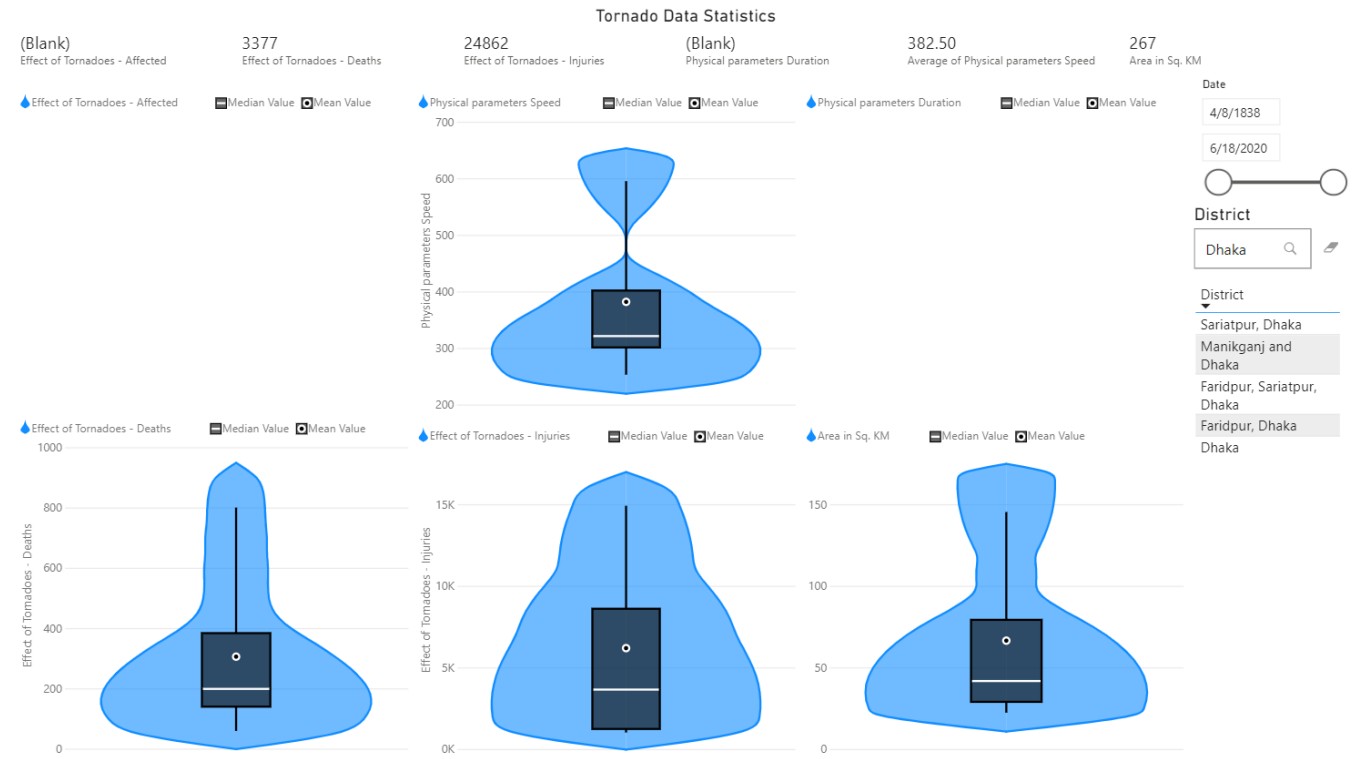

**Figure 6.** Data statistics, distribution & analysis of key numeric attributes for dhaka district (Case 2).

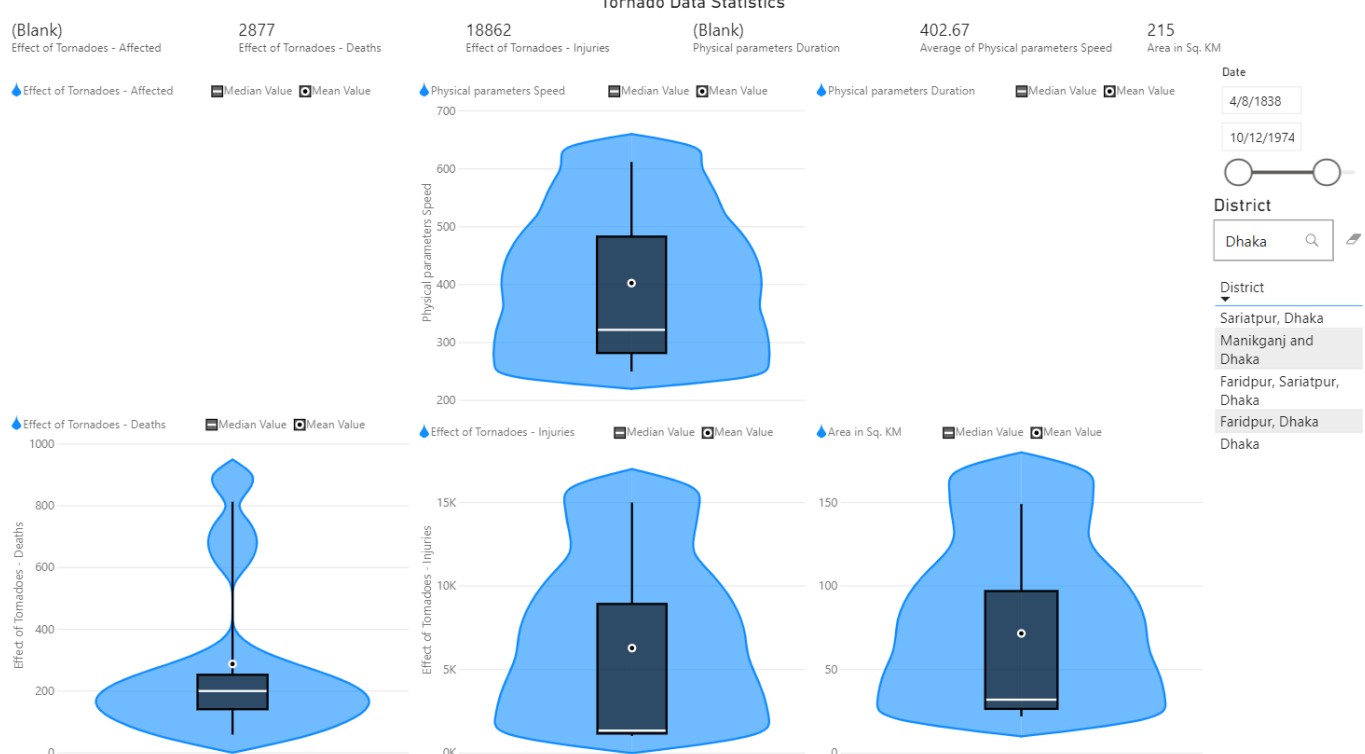

**Figure 7.** Data statistics, distribution & analysis of key numeric attributes for *Dhaka* district between dates *8 April 1838* to *12 October 1974* (Case 3).

*2.4. Analyzing the Tornado Data*

Within this research, two different types of dashboards were used: one for data exploration and another for data statistics. The data exploration report is presented in Figure 4. One the other hand, Figure 5 shows the dashboard for Data Statistics, Distribution & Analysis of Key Numeric Attributes. The six key numeric attributes for analysis are listed below:

1. Effect of Tornadoes-Affected
2. Effect of Tornadoes-Deaths
3. Effect of Tornadoes-Injuries
4. Physical parameters duration
5. Physical parameters speed
6. Area in Sq.KM

Violin Chart visualizations were used to analyze the data statistics and data distributions for six key numeric attributes. Within the scope of this paper, regression analysis was used to identify which feature attributes matter the most in Tornado-related fatalities. Our implementation of regression analysis automatically ranks the factors that matter, contrasts the relative importance of these factors, and displays them as key influencers for both categorical and numeric metrics. For numerical features, linear regression was performed using Microsoft's ML.Net's SDCA regression implementation [32]. Linear regression is one of the simplest machine learning algorithms that comes under supervised learning techniques and is used to solve regression problems. It is used to predict the continuous dependent variable with the help of independent variables. The goal of linear regression is to find the best-fit line that can accurately predict the output for the continuous dependent variable. By finding the best-fit line, the algorithm establishes a linear relationship between the dependent and independent variables in the form of Equation (1).

$$y = b_0 + b_1 x_1 + \varepsilon \tag{1}$$

On the other hand, for categorical features, logistic regression was performed using ML.Net's L-BFGS logistic regression [33,34]. Logistic regression is one of the most popular Machine Learning algorithms that use supervised learning techniques. It can also be used for classification and regression problems. Logistic regression was used to predict the categorical dependent variable with the help of independent variables using Equation (2).

$$\log[y/y - 1] = b_0 + b_1 x_1 + b_2 x_2 + \dots b_n x_n \tag{2}$$

The output of the logistic regression problem can only be between zero and one. Logistic regression can be used where the probabilities between two classes are required, such as whether it will rain today or not, either 0 or 1, true or false, etc.

Logistic regression without a threshold is a regression. However, by introducing a threshold within the process, transforms it into an efficient classifier. At the beginning we commence with the logistic or sigmoid function,

$$\sigma(t) = \frac{1}{1 + e^{-t}} \tag{3}$$

Which maps real number to interval (0, 1). Then, we proceed by defining the hypothesis function with,

$$h_\theta(x) = \sigma\left(\theta^T x\right) = \frac{1}{1 + e^{-\theta^T x}} \tag{4}$$

Classification decision is made on $y = 1$ when $h_\theta(x) \geq 0.5$ and $y = 0$ otherwise. The decision boundary is $\theta^T x = 0$. The cost function is represented by,

$$j(\theta) = \sum_{i=1}^{m} H(y^{(i)}, h_\theta\left(x^{(i)}\right)) \tag{5}$$

where $H(p, q)$ is the cross entropy of distribution $q$ relative to distribution $p$ and is given by,

$$H(p,q) = -\sum_i p_i \log q_i \tag{6}$$

In this case $y^{(i)} \in \{0,1\}$ so $p_1 = 1$ and $p_2 = 0$, therefore,

$$H\left(y^{(i)}, h_\theta(x^{(i)})\right) = -y^{(i)} \log h_\theta(x^{(i)}) - (1 - y^{(i)}) \log \left(1 - h_\theta\left(x^{(i)}\right)\right) \tag{7}$$

Similar to the selection of quadratic cost function in linear regression, the selection of this cost function is mainly driven by the fact that it is efficient and easy to implement as shown in,

$$\operatorname{grad} J(\theta) = \frac{\partial J(\theta)}{\partial \theta} = \begin{bmatrix} \frac{\partial}{\partial \theta_0} J(\theta) \\ \frac{\partial}{\partial \theta_1} J(\theta) \\ \cdot \\ \cdot \\ \cdot \\ \frac{\partial}{\partial \theta_n} J(\theta) \end{bmatrix} = X^T(h_\theta(X) - y) \tag{8}$$

Hence, gradient descent for logistic regression could be reflected with,

$$\theta(k + 1) = \theta(k) - s \operatorname{grad} J(\theta) \tag{9}$$

## 3. Results

Holding the mouse over each of the Violin charts representing the 6 key numeric attributes within the dashboard for data statistics, distribution & analysis reveal the detailed data statistics (as shown in Figure 8) within tooltips. These detailed data statistics include Number of Samples (containing non-blank values), Maximum, Minimum, Median, Mean and Standard Deviation. Apart from the data statistics, the violin chart also demonstrates the data distributions. For example, Figure 8 shows "Area in Sq.KM" attribute of Tornado data had only 15 samples. In other words, out of 119 records of our Tornado event database, only 15 samples contained values for "Area in Sq.KM" and the rest were all black values. The maximum value for "Area in Sq.KM" was 450 and the Minimum was 3. Median value was found to be 52 and the mean was 102.27. Lastly, Figure 8 shows standard deviation was 116.58. The blue colored area of the violin chart represents data distribution. As shown in Figure 8, most of the value within Area in Sq.KM were around 80 (since this is where the blue colored area is most expanded). Hence, Violin chart merges basic statistical information along with the data distribution (i.e., frequencies of data values on value ranges). The detailed data statistics for the 6 key numeric attributes are detailed in Table 2. It should be mentioned that these statistics are valid for the case 1, when no filters are applied.

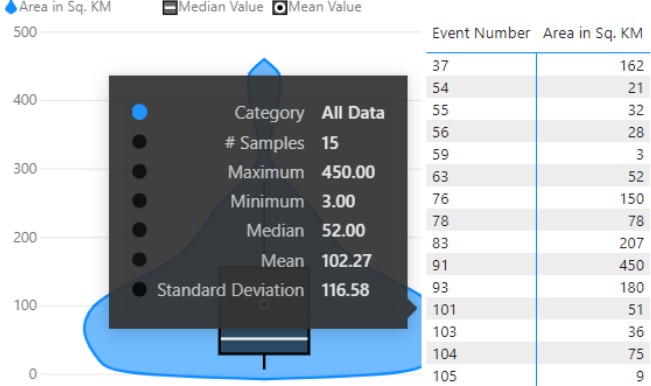

**Figure 8.** Tooltips of violin chart on attribute "*Area in Sq.KM*" revealing detailed data statistics.

**Table 2.** Case 1—data statistics for the 6 key numeric attributes without any filters.

| Attribute Name | Number of Samples | Maximum | Minimum | Median | Mean | Standard Deviation |
|---|---|---|---|---|---|---|
| 1. Effect of Tornadoes-Deaths | 103 | 922 | 1 | 28 | 84.81 | 146.77 |
| 2. Effect of Tornadoes-Injuries | 18 | 644 | 93 | 310 | 300 | 121.57 |
| 3. Effect of Tornadoes-Affected | 13 | 25 | 176 | 5.20 | 7.91 | 9.05 |
| 4. Areas in KM$^2$ | 15 | 450 | 3 | 52 | 102.27 | 116.58 |
| 5. Physical Parameters Duration | 4 | 8 | 2 | 4.5 | 4.75 | 2.50 |
| 6. Physical Parameters Speed | 18 | 644 | 93 | 310 | 300 | 121.57 |

When, we search only for district "Dhaka", then district filter is applied, and the data statistics are changed instantly for the filtered Tornado incidents that only took place in Dhaka. This is case 2 as shown in Figure 6. Table 3 shows the detailed data statistics for case 2.

**Table 3.** Case 2—data statistics for the 6 key numeric attributes with District = *Dhaka*.

| Attribute Name | Number of Samples | Maximum | Minimum | Median | Mean | Standard Deviation |
|---|---|---|---|---|---|---|
| 1. Effect of Tornadoes-Deaths | 11 | 922 | 46 | 200 | 307 | 277.11 |
| 2. Effect of Tornadoes-Injuries | 4 | 16.51 K | 1 K | 3.68 K | 6.22 K | 7.23 K |
| 3. Effect of Tornadoes-Affected | 0 | | | | | |
| 4. Areas in KM$^2$ | 4 | 162 | 21 | 42 | 66.75 | 64.78 |
| 5. Physical Parameters Duration | 0 | | | | | |
| 6. Physical Parameters Speed | 4 | 644 | 242 | 322 | 382.50 | 178.37 |

Lastly, in Case 3 two different types of filters were applied on the dashboard for Data Statistics, Distribution & Analysis of Key Numeric Attributes as shown in Figure 7. For this case, district is selected as *Dhaka* and the date ranges from *8 April 1838* to *12 October 1974*. The data statistics generated for case 3 is detailed in Table 4.

**Table 4.** Case 3—data statistics for the 6 key numeric attributes with District = *Dhaka* and Date = *8 April 1838* to *12 October 1974*.

| Attribute Name | Number of Samples | Maximum | Minimum | Median | Mean | Standard Deviation |
|---|---|---|---|---|---|---|
| 1. Effect of Tornadoes-Deaths | 10 | 922 | 46 | 200 | 287.70 | 284.20 |
| 2. Effect of Tornadoes-Injuries | 3 | 16.51 K | 1 K | 1.35 K | 6.29 K | 8.86 K |
| 3. Effect of Tornadoes-Affected | 0 | | | | | |
| 4. Areas in KM$^2$ | 3 | 162 | 21 | 32 | 71.67 | 78.42 |
| 5. Physical Parameters Duration | 0 | | | | | |

**Table 4.** *Cont*.

| Attribute Name | Number of Samples | Maximum | Minimum | Median | Mean | Standard Deviation |
|---|---|---|---|---|---|---|
| 6. Physical Parameters Speed | 3 | 644 | 242 | 322 | 402.67 | 212.79 |

Figures 9–11 demonstrate the how automated AI based regression analysis provided AI based knowledge discovery for the users of the designed system. Figure 9 represented a scenario, where the user selected a particular date range (between 2 January 1936 and 31 December 1993) and immediately our solution discovered the following drivers for Tornado associated deaths (in number of persons):

- When physical parameter speed goes up 123.16 (km/h), the average of Tornado associated death increases by 85.33.
- When Tornado injuries goes up 5446.62, the average of Tornado associated death increases by 47.61.

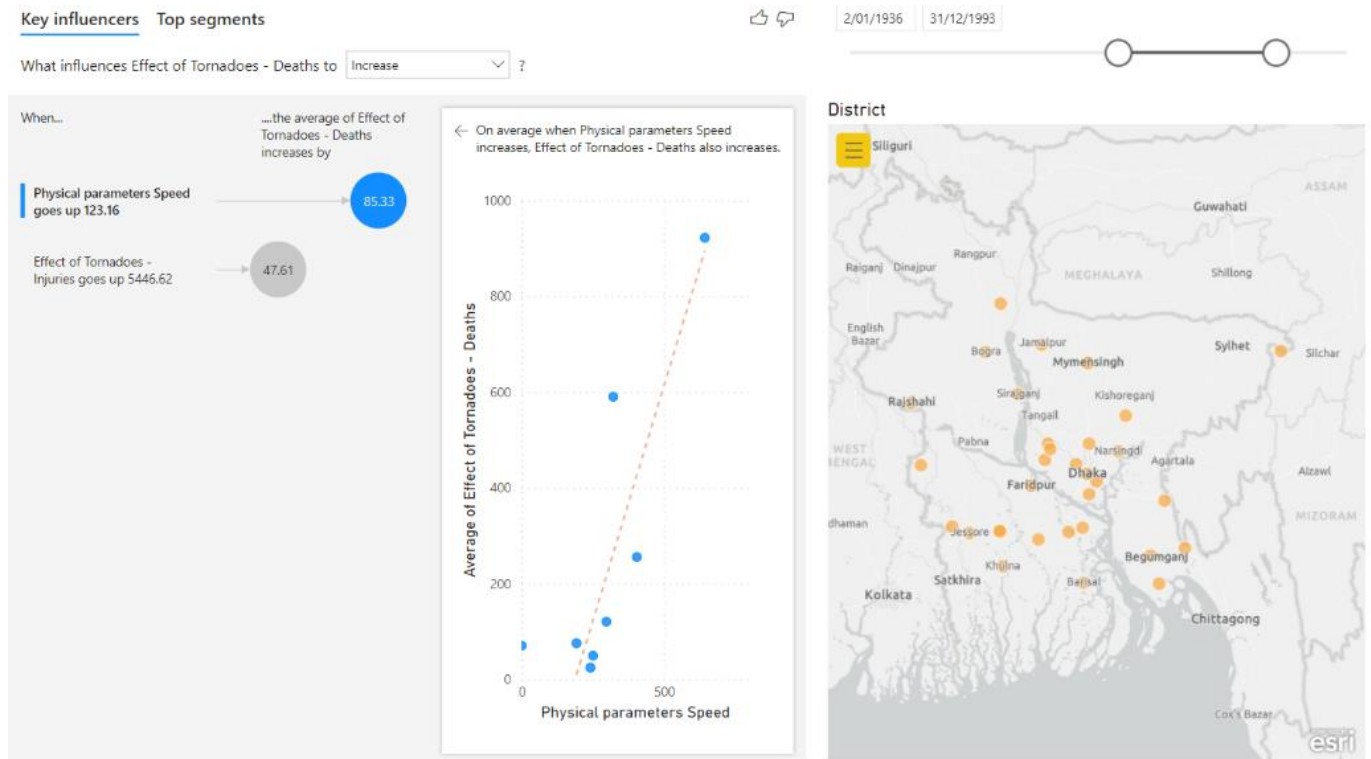

**Figure 9.** Regression algorithm automatically discovered factors relevant to Tornado associated deaths on filtered data between 2 January 1936 and 31 December 1993.

Similarly, Figure 10 represented a scenario, where the user selected a particular date range (between 31 March 1864 and 17 March 2002) and immediately our solution discovered the following drivers for Tornado associated deaths (in number of persons):

- When Tornado injuries goes up 6910.12, the average of Tornado associated death increases by 89.19.
- When physical parameter speed goes up 129.27, the average of Tornado associated death increases by 83.93.
- When Tornado affected goes up 6184.66, the average of Tornado associated death increases by 7.04.

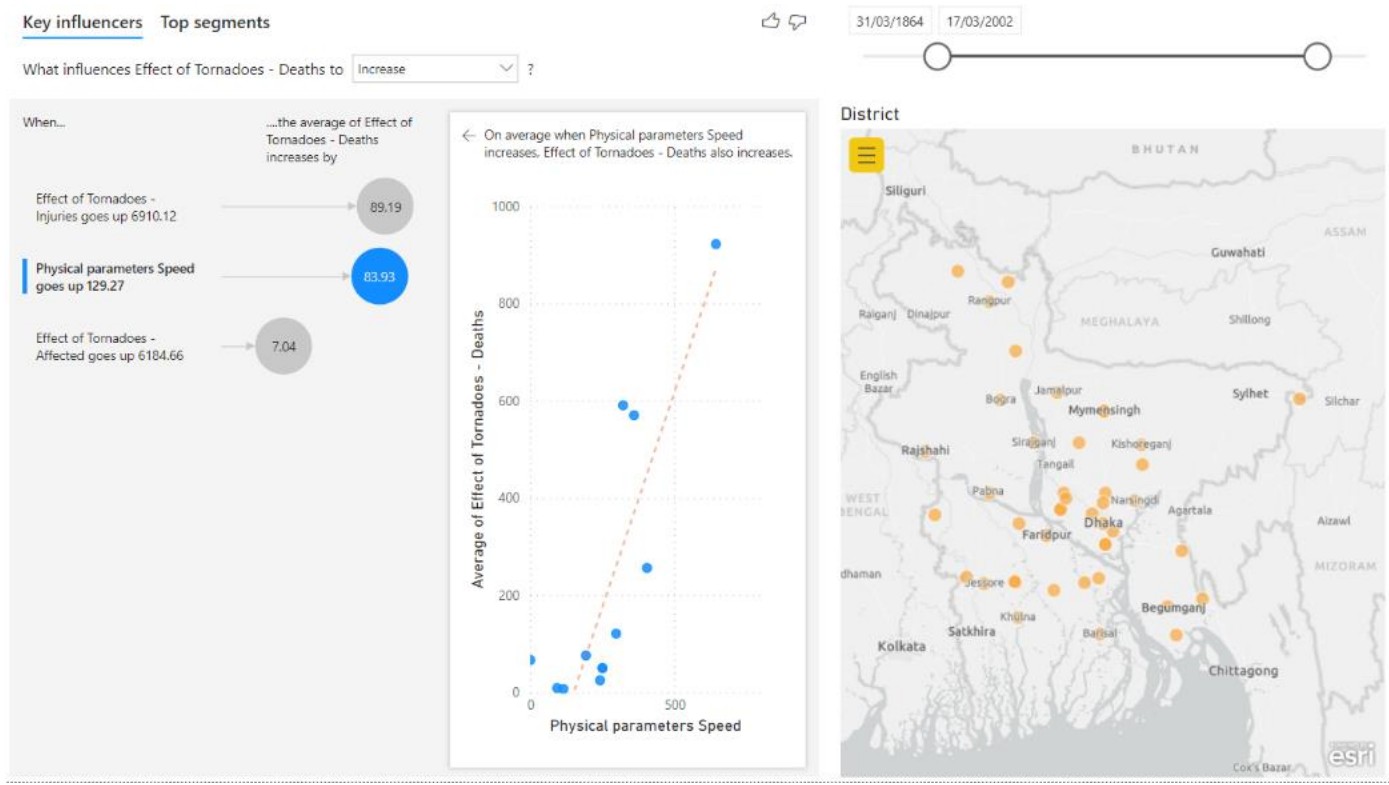

**Figure 10.** Regression algorithm automatically discovered factors relevant to Tornado associated deaths on filtered data between 31 March 1864 and 17 March 2002.

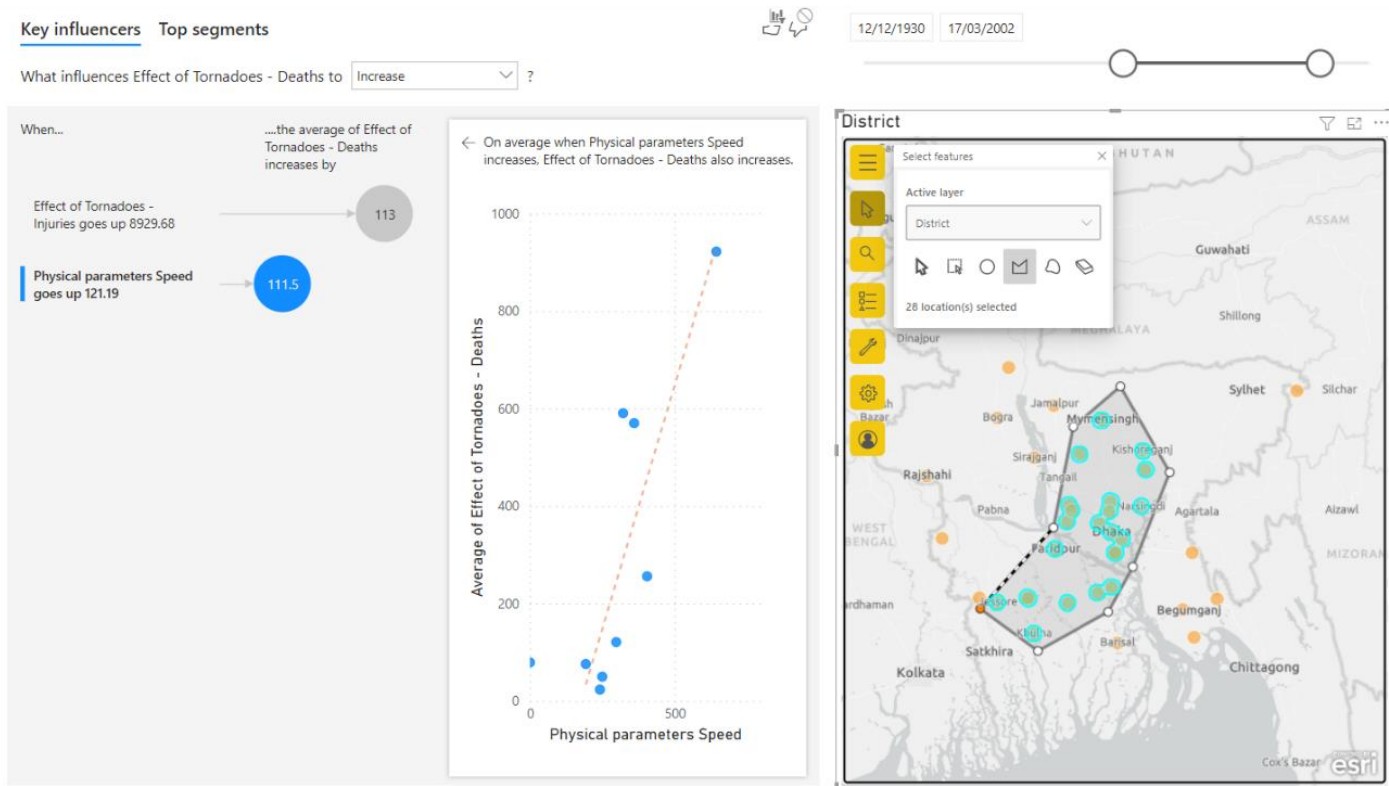

**Figure 11.** Regression algorithm automatically discovered factors relevant to Tornado associated deaths on filtered data between 12 December 1930 and 17 March 2002 and selected region.

Lastly, Figure 11 represented a scenario, where the user selected a particular date range (between 12 December 1930 and 17 March 2002) and selected a region inside the ESRI ArcGIS map; immediately our solution discovered the following drivers for Tornado associated deaths (in number of persons):

- When Tornado injuries goes up 8929.68, the average of Tornado associated death increases by 113.
- When physical parameter speed goes up 121.19, the average of Tornado associated death increases by 111.5.

## 4. Discussion

Existing studies of AI based Tornado Analysis as reported in [5–7] were driven by researchers and data scientists who understand AI Algorithms. In these studies, the authors manually inspected the data sources, decided on the suitable data cleansing, transformation, and AI algorithms to execute for harnessing the intended results. Finally, the results are explained by the authors. This mode of work, where a strategic decision maker needs to delegate the task to data scientists for modelling and reporting the result is not suitable for instant evidence-based decision making by the strategic decision maker.

In comparison to the existing literature in [5–7], this paper reported a unique mechanism where the strategic decision maker uses their own devices to select a particular scenario (i.e., through the presented decision support system) and immediately the best suited algorithms are automatically executed with outcomes reported in plain natural language. Using latest technological innovation in AI based Natural Language Processing (NLP) [35], the AI driven insights (i.e., the outcomes of executing AI algorithms) are explained in a language that is easily understandable by the strategic decision makers. In the previous section, we have seen (in Figures 9–11), explaining the results in natural languages like "*When physical parameter speed goes up 123.16 (km/h), the average of Tornado associated death increases by 85.33*" using NLP. Hence, the decision makers no longer need to rely on data scientists' explanation for linear or logistic regressions.

As shown in Figures 9–11, when the strategic decision maker selects a scenario, immediately the proposed solution filters the data and draws relationships with correlation coefficients on that filtered set of data. These figures demonstrate the AI based insights on discovering relationships of Effect of Tornadoes-Deaths with respect to Physical parameter speed, Effect of Tornadoes-Injuries, and Effect of Tornadoes-Affected. The proposed system hides all the complex AI based calculations from the user. Hence, this system is perfectly suitable for a strategic decision maker who does not have any background in Data Analysis, AI, ML, and Mathematical Modelling.

It should be mentioned that unlike statistical method of regression analysis (as shown in [36] to ascertain Tornado damage rating), the AI based regression analysis (demonstrated in this paper) is an end-to-end automated process (i.e., without the intervention of a data scientist), where plotting the relationship curves along with describing the relationship in natural language (using NLP) is a complete automated process.

While none of the existing research in Tornado [5–7,36] demonstrated applicability of their solutions through Mobile Applications (i.e., Mobile Apps) in Mobile or Tablet environment, the proposed solution was deployed in a range of devices including mobile, tablet, and traditional desktop machines. Figure 12 shows the app deployed through Android App in Samsung Galaxy 10 Lite mobile (equipment sourced from Dubai, UAE). On the other hand, Figure 13 shows the deployment through iOS App in Apple iPad 9th Generation. Applicability of the proposed solution in the widest range of platforms (i.e., iOS, Android, and Windows) makes it highly suitable for a strategic decision maker who can make instant decision using deployed app (i.e., Mobile Application) within mobile environment.

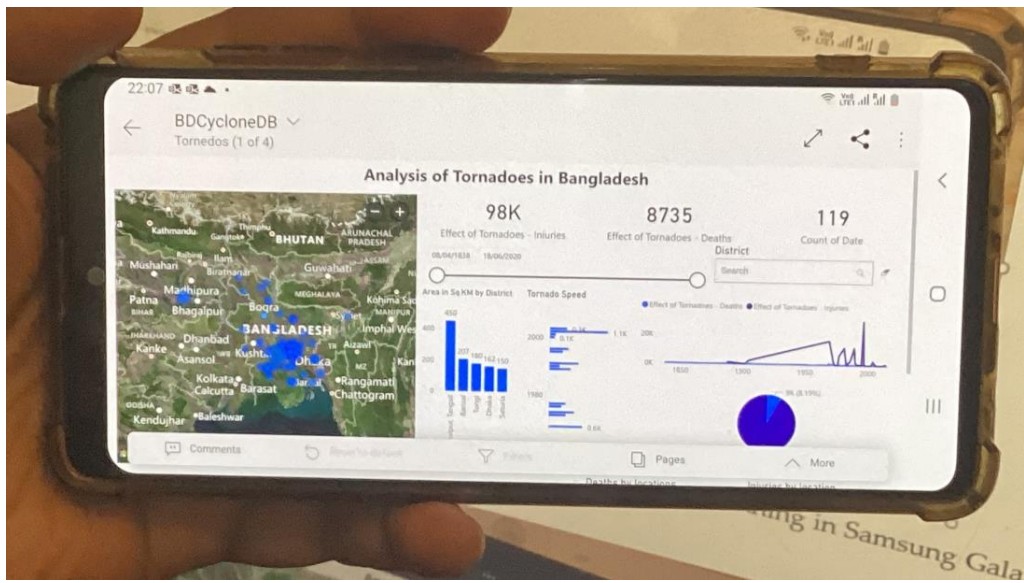

**Figure 12.** The proposed AI based Tornado analysis solution running in Samsung Galaxy Note 10 Lite mobile on Android 12.

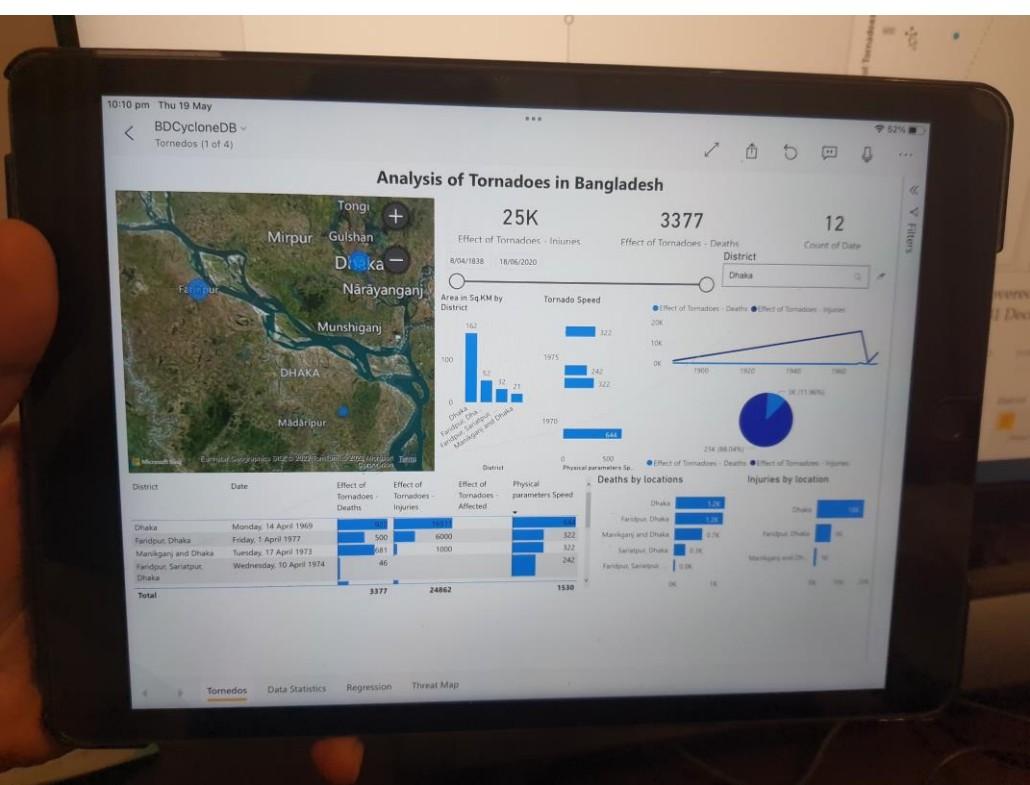

**Figure 13.** The proposed AI based Tornado analysis solution running in Apple iPad 9th Generation on iOS Version 15.

The user of our system can download the .pbix file from [18] and open using MS Power BI Desktop. MS Power BI is freely available for download from [19]. Typical user of this system would be town planners, policy makers, and disaster recovery strategist who is concerned with Tornadoes in Bangladesh. This system would allow the users to understand the characteristics of Tornadoes in a particular area and provide useful guidance for policy implementation for mitigating risks associated with Tornadoes in that area.

## 5. Conclusions

This paper reported a new dataset on Tornado incidence in Bangladesh covering the longest period (i.e., from 1838 till 2020) and made it publicly available for researchers, scientists, and disaster strategists. Based on this dataset, a new solution was developed that provided scenario-based Tornado insight using AI based regression analysis. Using this solution, disaster planners or strategists can obtain AI based insights on their mobile, tablet or smart devices. Following are the core contribution for this study:

- A new Tornado database is presented with 119 records starting from 1838 till 2020, causing 8735 deaths and 97,868 injuries leaving more than 102,776 people affected in Bangladesh.
- A novel AI based knowledge discovery solution was presented for disaster planners and strategist using AI based regression analysis. As shown in Figures 9–11, this novel solution immediately discovered the correlated factors with the correlation coefficients representing what drives Tornado associated deaths (e.g., speed of Tornado, injuries caused by Tornados etc.). It should be mentioned that the same techniques could be successfully applied to obtain the drivers for Tornado related injuries or other Tornado parameters.
- The designed system was deployed and tested on a range of devices on both iOS and Android platforms.

This experimentation was performed on a limited dataset containing only 119 records. Moreover, several data fields had empty values. As seen from Tables 1 and 2, 20% of district, 60% of Location of landfall, 87% of Area in Sq.KM, 85% of Physical parameters speed, 97% of Physical parameters Duration, 13% of Effect of Tornadoes-Death, 64% of Effect of Tornadoes-Injuries, 89% of Effect of Tornadoes-Affected, 87% of Effect of Tornadoes-Damage (in Million) and 85% of Key Comments were empty. Because of this limited record set and empty values within several Tornado attributes, the data analysis was performed on a very limited number of data points. The accuracy of the analysis generated by the presented system improves with the depth, breadth, and validity of the data. Furthermore, this research was conducted only using event-based historical data and did not consider Tornado data from the social media as reported in [2].

Therefore, in our future work, we will focus on a much larger set of data with a wider range of attributes containing lesser number of empty values. Moreover, in future, we endeavor to combine this Tornado event-based dataset with social media-based Tornado data (as reported in our recent research [2]) for performing deep learning. As shown in our recent studies, deep learning algorithms like Convolutional Neural Network (CNN) based anomaly detection can provide comprehensive understanding of disaster features [2,21,22].

**Author Contributions:** Conceptualization, F.S. and E.A.; methodology, F.S.; software, F.S.; validation, F.S. and M.A.; formal analysis, F.S. and E.A.; investigation, F.S., E.A. and M.A.; resources, F.S. and M.A.; data curation, F.S. and E.A.; writing—original draft preparation, F.S.; writing—review and editing, F.S. and E.A.; visualization, F.S.; funding acquisition, F.S. and M.A. All authors have read and agreed to the published version of the manuscript.

**Funding:** The authors would like to thank the Deanship of Scientific Research at Umm Al-Qura University for supporting this work by Grant Code: (22UQU4290525DSR02).

**Institutional Review Board Statement:** Not Applicable.

**Informed Consent Statement:** Not Applicable.

**Data Availability Statement:** Publicly available datasets were analyzed in this study. Microsoft Power BI tool was used to analyze the data with Microsoft Bing Maps, Violin Chart, Bar Chart, Pie Chart and Line Charts. Both Data (i.e., BDTornedoes.xlsx) and the MS Power BI Source file (i.e., BDTCycloneDB.pbix) This data can be found at [18].

**Acknowledgments:** The authors would like to thank Taufiqur Rahman, a development expert working for Federal Government, Canberra, ACT, Australia for his support during the development of dashboards for this study.

**Conflicts of Interest:** The authors declare no conflict of interest.

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
