# Peer review of "A New Decision Support System for Analyzing Factors of Tornado Related Deaths in Bangladesh"

_sustainability, doi:10.3390/su14106303_

Round 1

Reviewer 1 Report

This method could be applied to other countries and other disaster types. 

Author Response

We thank the honorable reviewer for the generous recommendation of accepting our paper.

Reviewer 2 Report

Dear author(s),

The methods, procedures, and steps of the research have been done logically. This version of the manuscript has been improved comparing the previous version. Most of the mentioned comments in that version have been considered and modified according to the suggestions and recommendations. The results are interesting and valuable now. Some minor corrections must be done before publishing in the respected journal. Therefore, according to the reviewer’s point of view, the following suggestions and recommendations can improve the quality of the text.

  • Those phrases listed as keywords should be used in the text. It is recommended to use the mentioned keywords in the text. Moreover, as a suggestion from the reviewer, “Logistic regression” seems a more appropriate keyword instead of “Regression” in the keywords list since in section 2.4, all the equations and explanations are evidence of using the “logistic regression model” to analyze the data.
  • Referring to some publication simultaneously should be mentioned as [1, 2] if there are two references, or [1-n] for more than 2 references that are listed consecutively. Therefore, referring in the second row of the Introduction section, line 29, should be corrected as the first-mentioned rule, and referring in line 35 should be corrected as [5-7]. It is highly recommended to check all the text and correct the other similar mistakes.
  • The word “Tornado” has been mentioned with a capital letter in starting and without it. Please see line 47 for instance. It is suggested to unify this word as “Tornado” or  “tornado” in whole the text.
  • If there was any other significant limitation except the limited number of the records, or any points to improve the precision of such a study in future studies, it is suggested to be mentioned in the Conclusion section.

Author Response

We are extremely pleased to know that 3 out of 4 reviewers have suggested our updated manuscript to be accepted for publication. We have accommodated all the minor changes suggested by the honorable reviewer. 

  • Those phrases listed as keywords should be used in the text. It is recommended to use the mentioned keywords in the text. Moreover, as a suggestion from the reviewer, “Logistic regression” seems a more appropriate keyword instead of “Regression” in the keywords list since in section 2.4, all the equations and explanations are evidence of using the “logistic regression model” to analyze the data.

We concur with the suggestion of the honorable reviewer. Hence, as per the suggestion, we have done the following changes:

  • Changed “Regression” to “Logistic Regression” within the keywords list
  • For using the keywords within the text, we updated the manuscript in a way that all the keywords are mentioned in the updated manuscript several times (please refer the following Table).

Keywords

Used within Manuscript

Logistic Regression

Line 59, 64, 81, 205, 206, 208, 212, 213, 215, 234, 321

AI based Tornado Analysis

Line 304, Line 351, Line 354

Decision Support System

Line 72, Line 313

Mobile Application

Line 336, Line 343

  • Referring to some publication simultaneously should be mentioned as [1, 2] if there are two references, or [1-n] for more than 2 references that are listed consecutively. Therefore, referring in the second row of the Introduction section, line 29, should be corrected as the first-mentioned rule, and referring in line 35 should be corrected as [5-7]. It is highly recommended to check all the text and correct the other similar mistakes.

Line 30 mentioned [1, 2] and line 36 / 40 mentioned [5-7] as suggested.

  • The word “Tornado” has been mentioned with a capital letter in starting and without it. Please see line 47 for instance. It is suggested to unify this word as “Tornado” or “tornado” in whole the text.

We concur this valuable suggestion. All the “Tornado” instances within the updated manuscript now start with capital T (i.e., there are no incidents of “tornado”).

  • If there was any other significant limitation except the limited number of the records, or any points to improve the precision of such a study in future studies, it is suggested to be mentioned in the Conclusion section.

We agree with this recommendation and accordingly, we have added the following limitation within the conclusion.

“Furthermore, this research was conducted only using event-based historical data and did not consider Tornado data from the social media as reported in [2].”

Subsequently, we mentioned that in our future work we will consider using aggregated dataset combining the event-based historical data as well as social media data.

Reviewer 3 Report

Only few small technical issues left. I appreciate the authors' effort to improve the manuscript.

1. Line 17, 50, 77, 362: Is the number format correct -> 1,02,776?

2. Fig. 4: Still "TornEdo speed" at the center of the figue. Also a bit more on the left - "Area in KM by District" - KM or km2?

3. Fig. 5-7: "Area in KM by District" - KM or km2?

4. Table 2-4, fig. 8: "Area in KM" - KM or km2?

Author Response

First of all, we would like to thank the honorable reviewer for accepting our manuscript for publication. We have accommodated all the minor changes suggested by the honorable reviewer. Our responses to the reviewers’ suggestions are now highlighted.  

  1. Line 17, 50, 77, 362: Is the number format correct -> 1,02,776?

For all these instances, we have now corrected this as 102,776.

  1. Fig. 4: Still "TornEdo speed" at the center of the figue. Also a bit more on the left - "Area in KM by District" - KM or km2?

We agree with suggestion and updated Figure 4 accordingly.

  1. Fig. 5-7: "Area in KM by District" - KM or km2?

We agree with suggestion and updated Figure 5 to Figure 7 accordingly.

  1. Table 2-4, fig. 8: "Area in KM" - KM or km2?

Table 2-4 as well as Figure 8 now updated as “Area in Sq. KM”

Reviewer 4 Report

I have no more comments on this article.

Author Response

We thank the honorable reviewer for the generous recommendation of accepting our paper.

This manuscript is a resubmission of an earlier submission. The following is a list of the peer review reports and author responses from that submission.

Round 1

Reviewer 1 Report

I have no comments on this article. Because it looks like a "pure" technical article without science.

The only comment is:

Please modify figure5, 6 & 7. Make sure the spell of the title is "Tornado Data Statistics" instead of "Tornedo". If authors want their system could be used by users.

Reviewer 2 Report

Dear author(s),

  • Those phrases listed as keywords must be used in the text. In this manuscript, for instance, “Disaster Management” has not been mentioned in the text at all. Moreover, the word “Tornado” has been used three times in the keywords list, which it is not necessary to mention that much. For example, “Decision Support System” and “Mobile Application” may be better to be used as such phrases. Please consider this issue and revise the keywords list precisely.
  • Referring to some publication simultaneously should be mentioned as [1, 2] if there are two references, or [1-n] for more than 2 references that are listed consecutively. Therefore, referring in the second row of the Introduction section, line 30, should be corrected as the first-mentioned rule, and referring in line 31 should be corrected as [1, 3-12]. It is highly recommended to check all the text and correct the other similar mistakes.
  • Lines 32 to 34 are the copy-paste of the lines 15 to 17. It is recommended to avoid such copy and pasting in the text. The information should be explained in the text with details and should be summarized in the Abstract section. It is suggested to check the rest of the manuscript to prevent this issue.
  • The first paragraph of the Introduction section is not really an intro for the introduction of the study. It contains summaries related to the data which should be mentioned after some studies from the literature or data sources. It is suggested to use definitions, and explain the concepts that have been covered in this study. Afterward, such data can be mentioned.
  • As mentioned in line 41, all the data are available via [13]. So, what is the necessity to mention those references in line 40, when all of them have been mentioned previously in line 31?!
  • The word “Tornado” has been mentioned with a capital letter in starting and without it. Please see lines 39 and 42 for instance. It is suggested to unify it in whole the text.
  • It is suggested to clarify the research gap, such as limitations and deficits of the literature, and the necessity of having this research complete the ligature gap.
  • Line 56: There is no section 3.4 in this manuscript! Check it, please.
  • Line 81: It has been mentioned: “the tornado data attributes can be categorized as … categorization …”. It is suggested to rewrite this sentence.
  • Line 136: “figure 5” should be written as “Figure 5”. Similarly, in line 150, where “Figure 5” has been mentioned as “Fig. 5”.Check the rest of the text to not have such types of mistakes.
  • Line 151: What words have “ML” has been a stand for? In the first time of using any acronym/abbreviation, the complete form must be mentioned. Please check the manuscript about preventing having such a problem.
  • The formula, mentioned in line 158 should have an equation number. Similarly, for the equation mentioned in line 164. The rest of the equation numbers should be updated.
  • In section 2.4, all the equations and explanations are evidence of using the “logistic regression model” to analyze the data. But, in the keywords list, line 25, “linear regression” has been mentioned. It is obvious that the respected authors have not used the linear regression method as the “Logistic regression” is not a linear one. It seems a modification is necessary about this issue.
  • It is suggested to make the rule of AI in the regression analysis clear.
  • The discussion section should be rewritten again, as some parts of it, are not related to the discussion. For instance, the sentence between lines 270 to 272 has been mentioned previously in the Abstract and Introduction sections. Basically, the contribution of the study is not related to the discussion part. Such information should be mentioned in the Introduction, literature, or in the conclusion sections.
  • In the Conclusion section, it is expected to explain the details of what had been done and found via this study. It is recommended to extend this section.
  • Referencing should be rechecked. For instance, reference [30] does not have the year that has been published.
  • There are many mistypes in this manuscript, as mentioned previously. It is recommended to check the whole text by a professional Editor.

Reviewer 3 Report

1. What is the research question/goal/thesis? What is the purpose ? These should be nicely stated at the beginning.

2. Line 17, 34, etc. - is the number correct?

3. Introduction:

-Background of the study is poor.

-In my opinion cited references have to be introduced better. Current format or maybe "a way" of referencing (like in lines 31, 40, 49) cannot be accepted. 11 references at the end of a single sentence look bad.

-3rd paragraph - it needs a better description of the way how decision making is supported. Cited literature include positions about landslides, Covid, social media policy, ECG, that are not directly related with the topic of this article. Almost a half of the references number is of the same author, which makes it look like improper self-citations.

4. What is surprising, in many places of the article the authors write about landslides, although in a way not related to the topic of the article. It looks like unwanted (forgotten to be deleted) remains from another article. E.g.: line 60 (wrong caption), 88, 89, 90 (again caption + figure content "Other - Location of Landfall"). If otherwise, please explain how the landfall and landslides are related to the current topic of the article.

5. Table 1:

-Table 1 is insufficiently introduced.

-Some not so obvious captions should be explained as well, especially these under Attribute Statistics. The same applies to attributes.

-Basically the method of gathering Attribute statistics is missing! How the numbers were discovered/set. Maybe a short example? Any problems when gathering Attribute statistics? What is the meaning of the numbers? How to understand these numbers?

-Attribute no. 4 "Location of landfall" - again landfall - is it correct?

- Attribute no. 5 "Area in KM" - in KM or square km? what is the unit?

-Attribute no. 6 "... speed" - again, what unit? km/h?

-Attr. no. 7. "... duration" - again, what unit? day, hour, minute?

-Attr. no. 8. "... death" - but who/what was counted as death? person? animal?

-Attr. no. 9. "... injuries" - people injured or injuries (many injuries counted for 1 person e.g. head, limbs, etc.)?

-Attr. no. 10. "...affected" - explain who is affected - persons/buildings?

-Attr. no. 11. "...damage" - in Million what? beans? bitcoins? USD? currency? ruined buildings? killed house animals?

-Attr. no. 12. "Key comment" - what is a key comment? Example?

6. Figure 3 - "Tornado Data" - injuries missing on the list. Death is mentioned 2 times.

7. Figure 4. TornEdo! speed unit 0.6K. Explain what happened.

8. fig. 5-7.

- TornEdo ! mistake in the captions (at the top of the figures).

- Moreover, is the speed summed up? 1530 units ?

-What about case 1? It's absence is ok, but it is surprising. Maybe short notice in the text?

-Case 2 and 3 differ only in time span. What is the purpose of such cases?

9. Line 150-151 - please write, how it is shown in Fig.5.

10. Fig. 8. Information in the table is not particularly useful for an app user, as it does not facilitate the recognition of a particular event (tornado occurrence).

11. Lines 229, 230, 236, 238, 240 and similar - increases by what? units? persons?

12. The AI implementation is poorly addressed, especially in context of the mobile app.

13. Why figures 9-11 are worth publishing? Why not only 1-2 of them? Are they necessary? Why are they important - justify.

14. Lines 266-267 - something is missing in the sentence.

15. Line 270 - Why only 119 records were identified?

16. Lines 273-276 - Why only deaths?

17. References should be improved. Include some new refs and remove refs that are not directly related to the topic of the article. Also, proportion of the authors' articles should be addressed.